

# Genome sizes and repeatome evolution in zoantharians (Cnidaria: Hexacorallia: Zoantharia)

Chloé Julie Loïs Fourreau[1], Hiroki Kise[1,2], Mylena Daiana Santander[3], Stacy Pirro[4], Maximiliano M. Maronna[1,5], Angelo Poliseno[1], Maria E.A. Santos[1,6] and James Davis Reimer[1,7]

[1] Molecular Invertebrate Systematics and Ecology (MISE) Lab, Graduate School of Engineering and Science, University of the Ryukyus, Nishihara, Okinawa, Japan
[2] AIST Tsukuba Central, Geological Survey of Japan, National Institute of Advanced Industrial Science and Technology, Tsukuba, Ibaraki, Japan
[3] Departamento de Genética e Biologia Evolutiva, Instituto de Biociências, Universidade de São Paulo, São Paulo, Brazil
[4] Iridian Genomes, Bethesda, United States of America
[5] Faculdade de Ciências, Universidade Estadual Paulista (UNESP), Bauru, Brazil
[6] Okinawa Institute of Science and Technology, Onna, Okinawa, Japan
[7] Tropical Biosphere Research Center, University of the Ryukyus, Nishihara, Okinawa, United States of America

Corresponding author
Chloé Julie Loïs Fourreau, chloisf@gmail.com

## ABSTRACT

Across eukaryotes, large variations of genome sizes have been observed even between closely related species. Transposable elements as part of the repeated DNA have been proposed and confirmed as one of the most important contributors to genome size variation. However, the evolutionary implications of genome size variation and transposable element dynamics are not well understood. Together with phenotypic traits, they are commonly referred to as the "C-value enigma". The order Zoantharia are benthic cnidarians found from intertidal zones to the deep sea, and some species are particularly abundant in coral reefs. Despite their high ecological relevance, zoantharians have yet to be largely studied from the genomic point of view. This study aims at investigating the role of the repeatome (total content of repeated elements) in genome size variations across the order Zoantharia. To this end, whole-genomes of 32 zoantharian species representing five families were sequenced. Genome sizes were estimated and the abundances of different repeat classes were assessed. In addition, the repeat overlap between species was assessed by a sequence clustering method. The genome sizes in the dataset varied up to 2.4 fold magnitude. Significant correlations between genome size, repeated DNA content and transposable elements, respectively (Pearson's correlation test $R^2 = 0.47$, $p = 0.0016$; $R^2 = 0.22$, $p = 0.05$) were found, suggesting their involvement in the dynamics of genome expansion and reduction. In all species, long interspersed nuclear elements and DNA transposons were the most abundant identified elements. These transposable elements also appeared to have had a recent expansion event. This was in contrast to the comparative clustering analysis which revealed species-specific patterns of satellite elements' amplification. In summary, the genome sizes of zoantharians likely result from the complex dynamics of repeated elements. Finally, the majority of repeated elements (up to 70%) could not be annotated to a known repeat class, highlighting the need to further investigate

non-model cnidarian genomes. More research is needed to understand how repeated DNA dynamics relate to zoantharian evolution and their biology.

## INTRODUCTION

The C-value, the size of a species' haploid genome, has long been noticed to exhibit considerable variations that do not correlate with a given species expected complexity. This discrepancy has been referred to the "C-value paradox", highlighting the confusion regarding this pattern (*Thomas, 1971*; *Elliott & Gregory, 2015*). Indeed, despite genome sizes being in most cases relatively constant within species (*Swift, 1950*; *Gregory & Johnston, 2008*; *Dai et al., 2022*), intraspecific variation is well recognized (*Bonnivard et al., 2009*; *Stelzer, Pichler & Hatheuer, 2021a*) and large variations exist between closely related species (*Yuan et al., 2018*; *Wong et al., 2019*; *Becking, Gilbert & Cordaux, 2020*; *Shah, Hoffman & Schielzeth, 2020*; *Paule et al., 2021*). The discrepancies between genome size, phenotype complexity and genomic content was reframed by the discovery of large amounts of repetitive DNA in genomes (*Gregory, 2005*). In particular, the relative content of transposable elements has been found to explain dynamics of interspecific genome sizes variations in many groups (*Kapusta, Suh & Feschotte, 2017*; *Lee & Kim, 2014*; *Wong et al., 2019*; *Becking, Gilbert & Cordaux, 2020*; *Lehmann et al., 2021*; *Meyer et al., 2021*). However, this finding raised even more questions regarding the impact of these repetitive elements (including both protein-coding and non-coding sequences) on evolutionary dynamics: type of elements involved, mechanisms (*e.g.*, amplification), historical processes (gain or loss of DNA content), and how repeats may relate to organismal and ecological traits. The set of questions that have risen from deciphering the "C-value paradox" are now collectively referred to as the "C-value enigma" (*Gregory, 2005*).

The development of next-generation sequencing along with tools dedicated to the annotation of specific repeated elements has allowed to describe and identify in detail various classes of genome repetitive elements, several of which are potentially involved in genome size variations. Currently, they are classified into two large groups based on their potential for mobility; tandem repeats and transposable elements. Tandem repeats include satellites, microsatellites, and rDNA (*Bourque et al., 2018*). On the other hand, transposable elements (TEs) can move within a genome and are distinguished into two classes based on their transposition mechanisms (*Wicker et al., 2007*). Class I TEs, also known as retrotransposons, insert themselves by reverse transcription; they include long terminal repeats (LTRs), long interspersed nuclear elements (LINEs), and short interspersed nuclear elements (SINEs). Transposable elements of class II encode for a transposase, an enzyme that performs transposition. These elements include Helitrons, Maverick and other DNA transposons subcategories (*Wicker et al., 2007*). LTRs have been shown to largely impact genome sizes in plants (*Dai et al., 2022*) and salamanders (*Sun et al., 2012*),

LINEs have been involved in *Hydra* (*Wong et al., 2019*) and in the giant lungfish genome (*Meyer et al., 2021*), while the implicated group of elements were SINEs in larvaceans (*Naville et al., 2019*) and DNA transposons in fish (*Lehmann et al., 2021*).

Different effects on the genomes may be considered along with the different groups of transposable elements involved. Repeated elements have been referred to as "junk DNA" and were initially thought to be neutral with regards to genome evolution. However, their dynamics can have large implications on the genome and species biology. For example, TEs can have adverse effects on their host by causing cancer (*Bourque et al., 2018*), including transmissible cancers through horizontal transfers in the marine environment (*Metzger et al., 2018*). Furthermore, TEs can lead to sequence polymorphism and gene diversification through genomic rearrangements and mediation of gene expression. As examples of this, transposable elements have promoted the diversification of opsins in the amphioxus genome (*Pantzartzi, Pergner & Kozmik, 2018*), and a TE insertion event gave rise to the dark morphotype of the peppered moth (*Van't Hof et al., 2016*). Finally, TEs have been associated with hybrid defects, and are thus potentially involved in the speciation process (*Serrato-Capuchina & Matute, 2018*). For all these reasons, repeated elements are relevant to the understanding of species biology and evolution. Considering the variety of elements involved in different taxa and their large array of potential implications, a better understanding of repeated elements and genome sizes requires further research on understudied groups (*Elliott & Gregory, 2015*). *Hotaling, Kelley & Frandsen (2021)* highlighted important taxonomic biases in genome sequencing projects, showing large research bias in favor of vertebrates. This is also true for the study of repeatome and genome sizes. Many groups still lack basic genome size information, as seen in the genome size database (*Gregory, 2023*: https://www.genomesize.com/), where most groups of invertebrates have less than 100 recorded genome sizes, whereas fish, insects and mammals have several hundreds to greater than a thousand records. In phylum Cnidaria, the first study documenting genome sizes across a wide taxonomic scope was published in 2017 by *Adachi et al.* While most cnidarians seem to have relatively small genomes (*e.g.*, mean C values: 0.70 pg for Anthozoa, 0.46 pg for Scyphozoa, and 1.20 pg for Hydrozoa) compared to other metazoans, there is a >13-fold variation in their genome diversity, (from 0.26 pg in scyphozoan *Sanderia malayensis* to 3.56 pg in hydrozoan *Agalma elegans*; *Adachi et al., 2017*). However, more research is needed to fully understand the scope and diversity of genome size variation in Cnidaria. Zoantharians represent one of the several taxa within the phylum for which no estimates of genome sizes have yet been published.

The order Zoantharia Rafinesque, 1815 is considered the earliest branching hexacorallian group (*Quattrini et al., 2020*) and their study harbors important implications for the evolution of cnidarian traits including skeleton production (*Quattrini et al., 2020*), symbioses, coloniality, and development (*Hirose et al., 2011*). Zoantharians are extensively distributed in subtropical and tropical oceanic regions and inhabit intertidal zones to the deep sea (*Santos et al., 2019*) and, in certain environments, can be dominant (*Yang et al., 2013*). In suborder Brachycnemina, most species establish symbiosis with photosynthetic dinoflagellates of the family Symbiodiniaceae, and azooxanthellate species (*i.e.,* that do not host Symbiodiniaceae) are thought to have lost this relationship (*Irei, Sinniger &*

*Reimer, 2015*). On the other hand, zoantharians of the suborder Macrocnemina are usually azooxanthellate, and epizoic on a range of marine invertebrates, including sponges, hermit crabs, molluscs, annelids, urchins, and several different groups of anthozoans (*Kise, Maeda & Reimer, 2019*). In addition, some species of zoantharians are known to produce palytoxin, one of the most potent toxic compounds known from the marine environment (*Aratake et al., 2016*), and present potential therapeutical applications. The phylogenetic relationships of zoantharians are currently debated and have been the focus of a few phylogenomic reconstructions; examples include a detailed phylogeny of genus *Palythoa* from ezRAD (*Dudoit et al., 2021*), the placement of Zoantharia within Cnidaria from ultra-conserved elements (*Quattrini et al., 2020*), and the phylogeny of Zoantharia from mitochondrial genome datasets (*Poliseno et al., 2020*). Some of these phylogenies (*Poliseno et al., 2020*; *Quattrini et al., 2020*) together with previous single marker phylogenetic results indicate that the taxonomy of zoantharians should be revised, since Brachycnemina is nested within Macrocnemina (*Sinniger et al., 2005*). Therefore, genomic data produced on zoantharians so far has mainly been employed to investigate phylogenetic relationships. However and in spite of high relevance of zoantharians in terms of evolution, ecology and biochemical potential, basic understanding of their genomes are lacking, including a lack of information about their genome sizes and repetitive element content.

To fill this gap we investigated genomic data of 32 species of zoantharians, spanning 10 genera of the order and five out of nine families. We present newly sequenced data for 17 of those species. From this recent and mostly unexplored molecular resource, we aimed to (1) expand present mitochondrial data *via* increased taxon sampling to test the current view of zoantharian phylogeny, (2) provide baseline data on zoantharian genomes with regards to genome sizes and repeatomes, and (3) assess the relative importance of different repeated DNA classes in genome size evolution in the order.

## MATERIAL AND METHODS

### Sampling and sequencing

Thirty-two specimens of zoantharians were gathered from SCUBA diving, scientific deep-sea expeditions, and museum collections between 1982 and 2019, from the Pacific Ocean, the Caribbean Sea and the South African coast of the Indian Ocean. These specimens were fixed in 99% ethanol and kept at −20 °C before 30 of them were sent to Iridian Genomes (Bethesda, USA) for whole-genome sequencing. DNA was extracted using the Qiagen DNeasy kit following manual's instructions. The sequencing platform, Illumina Hi-Seq X-Ten, generated approximately 60 million paired-end reads of a size of 150 bp per specimen. Genome data for 11 brachycnemic zoantharian specimens (*Santos et al., 2023*) and the five *Epizoanthus* species in the scope of the present paper have been already presented (*Kise, Reimer & Pirro, 2023*). In the case of the sample of *Palythoa mizigama*, DNA was extracted by CTAB-based protocol and sequenced at the NovoGene Hong Kong facility using the Illumina HiSeq X Platform (NEBNext® DNA Library Prep Kit was used for library construction (350 pb insert size, 150 pb read length), including size selection and PCR-enrichment, with a total input amount of 1.0 μg DNA). For the whole genome

sequencing of *P. tuberculosa* ∼1 μg of genomic DNA was sent to Admera Health (South Plainfield, NY). Genomic library was prepared using a Kapa® HyperPrep kit (Roche) and it was sequenced on an Illumina Hi-Seq X platform using a 150 pair-end chemistry.

The sequencing experimental data are available on the Sequence Read Archive (SRA) with accession numbers and corresponding information on the specimen collection are listed in Table S1. All SRA paired-end reads were downloaded onto the National Institute of Genetics Supercomputer Cluster (https://sc.ddbj.nig.ac.jp/en) to proceed with subsequent bioinformatic analyses. Before any analyses, the samples were quality-checked using FastQC v. 0.11.9 (*Andrews, 2010*). Prior to the following analyses, paired-end reads adapter sequences were removed in Trimmomatic v. 0.39 with default parameters (*Bolger, Lohse & Usadel, 2014*).

## Mitochondrial genome assembly and phylogeny

Mitochondrial genomes (mtDNA) were assembled *de-novo* with NOVOPlasty v. 3.8.3 (*Dierckxsens, Mardulyn & Smits, 2017*), with a *k*-mer size comprised from 29 to 33. A partial *COI* sequence (∼780 bp) from *Palythoa tuberculosa* (GenBank accession number: MH013403) was chosen as seed for the assembly of the majority of the samples, yet for others we used the sequence of phylogenetically close species (Accession numbers of sequences used as seeds are listed in Table S2). Although the assembly was performed *de novo*, the input of a reference genome facilitates the process, hence the mitogenome of *Palythoa heliodiscus* was used (*Chi & Johansen, 2017*; NC035579). To identify the gene composition and order, mitochondrial genomes were circularized and annotated in Geneious v.8.1.9. (*Kearse et al., 2012*). Annotation was done using the Predict and Annotate tool by comparing mitogenomes with a reference mitogenome annotation of *Palythoa heliodiscus* (NC035579) and other zoantharian mt-genomes from *Poliseno et al. (2020)*. Protein-coding sequences with >75% similarity to a gene in the reference were assigned to the corresponding gene. This same method was used to annotate tRNA and rRNA genes.

To infer the evolutionary relationships of zoantharians, phylogenetic trees were inferred based on mitochondrial protein coding genes. Thirteen genes (*COI, COII, COIII, CYTB, ATP6, ATP8, NAD1, NAD2, NAD3, NAD4, NAD4L, NAD5, NAD6*) were retrieved from each genome and aligned individually with MUSCLE (*Edgar, 2004*). Additional mitogenomes available from the literature and incorporated in the dataset are listed in Table S3 along with two antipatharians mitogenome assemblies that were used as outgroups in the phylogenetic trees (*Kayal et al., 2018*, Kwak, Choi *et* Hwang, unpublished). The thirteen alignments were concatenated in Sequence Matrix v.1.8 (*Vaidya, Lohman & Meier, 2011*), resulting in 11,933 bp matrix. The best fitting evolutionary model of each gene was assessed with MEGA X (*Kumar, Stecher & Tamura, 2016*) using the AIC criterion (*Akaike, 1973*). Sequence evolution models were HKY+G+I (*Hasegawa, Kishino & Yano, 1985*) for *ND5* and *ND4L*, GTR+G+I (*Tavaré, 1986*) for *ND1, ND2, ND3, ND4, ND6, CYTB, COIII, COI, ATP6*, and T92+G+I (*Tamura, 1992*) was the best fitting model for *COII* and *ATP8*. Because T92+G+I was not available in MrBayes nor raxml-ng, the second-best fitting model was employed for these two genes, which was in both cases HKY+G+I.
Based on the concatenated alignment, phylogenetic trees were computed following the maximum-likelihood method (ML) in RaxML-NG using the command—all (*Kozlov et al., 2019*), which comprises of an initial tree search step and a non-parametric bootstrapping step with node support estimated by 1,000 replicates. Furthermore, a Bayesian phylogenetic tree was inferred with MrBayes v.3.2.7 (*Ronquist & Huelsenbeck, 2003*). Each Monte Carlo Markov Chain (MCMC) was sampled every 1,000 steps during 10 $10^6$ generation cycles, and the first 25% of the trees were discarded as burn-in. Tree node parametric support was evaluated with the Bayesian posterior probabilities calculated during the analysis. For both the ML and the Bayesian tree computations, partitions were set with the corresponding sequence evolution model of each gene.

## Comparative genomic analyses
### Genome sizes
To estimate genome sizes, the $k$-mer frequencies of previously trimmed reads were counted in Jellyfish (*Marçais & Kingsford, 2011*) with the default $k$-mer size of 21. The resulting histograms were then input in GenomeScope (*Vurture et al., 2017*), which estimates genome size based on the distribution of a given $k$-mer size.

## Abundance and annotation of repeat classes
The pipeline dnaPipeTE v.1.3.1 (*Goubert et al., 2015*) was employed to assemble, annotate and estimate the abundance of repeated elements in each zoantharian genomic dataset. This software uses low coverage read samples to assemble representative contigs of repeats with Trinity v.2.5.1 (*Grabherr et al., 2011*) and then annotates the resulting contigs with Repeatmasker (*Smit, Hubley & Green, 2013-2015*, RepeatMasker Open 4.0.7) and RepBase (*Bao, Kojima & Kohany, 2015*). The dnaPipeTE pipeline also estimates repeat abundances and the divergence of repeat copies to the assembled contigs *via* blastn (*Altschul et al., 1990*). Both pieces of information are then used to estimate the landscape distribution of repeated elements, as a proxy of their relative age. To ensure the sampling of repeated elements, reads were trimmed and removed with stricter parameters than the default Trimmomatic command. The chosen parameters demanded a minimum read length of 140 bp instead of the default 36 bp (MINLEN:140), as well as an average quality below 20, instead of the default 15 (SLIDNGWINDOW:4:40). To avoid misrepresenting the repeat composition, non-repeat sequences with high coverage must be filtered out of the dataset (*Goubert et al., 2015*). The mitochondrial genomes previously assembled were removed from the trimmed reads using the script bbsplit.sh from bbmap package (*Bushnell, 2014*). To establish whether Symbiodiniaceae DNA contamination affected the results of genome sizes estimation and repeated elements analysis, contamination was checked using FastQScreen v.0.14.1 (*Wingett & Andrews, 2018*). For each species' dataset, approximately 100,000 reads (strictly trimmed, with mitochondrial genomes removed read datasets) were sampled and mapped against available genome assemblies of Symbiodiniaceae. The assemblies included one *Cladocopium* sp. and one *Symbiodinium* sp. Clade A3 generated by *Shoguchi et al. (2018)*, as well as *Breviolum minutum* (*Shoguchi et al., 2013*), *Durusdinium trenchii* (*Shoguchi et al., 2021*), *Symbiodinium microadriaticum* (*Aranda et al., 2016*), *Cladocopium proliferum* (*Liu et al., 2018*; genome presented as *C. goreaui* but species later named

*C. proliferum* in *Butler et al., 2023*), *Fugacium kawagutii* (*Liu et al., 2018*), and the human genome assembly GRCh37 (*Church et al., 2011*). FastQscreen outputs the number of reads from the dataset that mapped to reference genomes. When reads map several times over multiple genomes, this may indicate that instead of true contamination, very similar regions exist between some the reference assemblies and the target dataset. On the other hand, a large amount of reads mapping to a single genome must be interpreted as a serious case of contamination to consider (*Wingett & Andrews, 2018*).

To produce comparable estimates of repeated elements between species, the "fixed read sampling size" method was used (as opposed to using genome coverage). To determine the appropriate number of reads to sample, tests were run by providing genome sizes, and with dnaPipeTE coverage options of 0.1, 0.2, 0.25, 0.3, 0.4, and 0.5 fold for two of the datasets with largest genomes (*Palythoa tuberculosa* and *Umimayanthus chanpuru*) and one of the smallest (*Hydrozoanthus tunicans*). The resulting Trinity assemblies of annotated and unannotated contigs (annoted.fasta, unannoted.fasta, Trinity.fasta output files) were evaluated with the L50 metric using the bbtools script stats.sh (Table S4). Based on this, the optimal sample size (number of reads) was assessed using the formula $C = (N*L)/G$ with C the coverage, N the number of reads, L the read length (150 bp) and G the genome size. To determine N, C was set as 0.4 based on the results of dnaPipeTE test runs (Table S4). To ensure that all datasets were sufficiently sampled, G was input as the smallest genome size recovered, from *Palythoa mizigama* ($G = 286,669,957$ bp). Based on this calculation, the read sampling size was fixed to 764,453 for all species. In addition, the minimum size of contig to be included was set to 400 bp. Finally, the output files "Counts.txt" and "reads_landscape" of dnaPipeTE analysis, containing counts of each annotated repeat class, per species, were employed for statistical analyses.

## Repeat clustering and comparative composition between species

To analyze whether sequences of different classes of repeated DNA were shared between zoantharian species, a comparative analysis was performed with RepeatExplorer2 (*Novák, Neumann & Macas, 2020*). This pipeline allows the clustering, quantification and annotation of repeats from unassembled short reads, on the web interface Galaxy. It was employed in comparative mode for the 18 species with available genome size information. Pre-processing was performed on RepeatExplorer2 as described in Protocol 2 of the pipeline manual (*Novák, Neumann & Macas, 2020*). The clustering of the reads was performed in comparative mode using the RepeatExplorer database in Metazoa version 3.0 and default parameters. In this process, RepeatExplorer2 performs the clustering of the reads regardless of the species they belong to. Therefore, similar reads of different species clustered together, representing groups of repeated elements that are shared between different species. On the other hand, clusters that were composed of reads from a single species were considered specific repeats. The RepeatExplorer2 clustering outputs a list of superclusters along with their annotation. Because of conflicts during the annotation process, each supercluster annotation was reviewed and manually corrected as advised by *Novák, Neumann & Macas (2020)*. Clusters that could not be assigned to a repeat type were viewed in tablet (*Milne et al., 2013*) and the contig with the most important number

of reads was inspected. When the reads at the tip portions of the contig showed high polymorphism, the cluster was considered a mobile element (*Novák, Neumann & Macas, 2020*), as this structure represents several different insertion sites of a transposon. Clusters that did not display such pattern remained unannotated. Finally, in order to visualize clusters that were shared or not shared between species, the corrected version of the cluster annotation file was input in RepeatExplorer visualizing tool.

## Statistical analyses and visualization

To be able to relate evolutionary history with repeat abundance and genome sizes, a cladogram was drawn based on the topology of phylogenetic trees computed with the mitochondrial datasets, pruning the branches of specimens without genome size data in TreeViewer v.2.0.1 (*Bianchini & Sánchez-Baracaldo, 2023*). As the mitogenome of *Umimayanthus chanpuru* could not be reconstructed, this species was placed on the cladogram with *Umimayanthus nakama*, based on phylogenetic reconstructions from the literature (*Montenegro, Sinniger & Reimer, 2015*). Results were visualized with the R package ggtree (*Yu, 2020*).

To evaluate whether genome sizes were correlated to total repeated elements or transposable elements, a regression analysis was performed with the lm function in R (*R Core Team, 2021*). Only the transposable elements classified as such were included into the analyses. Normality of residuals distribution was assessed with a Shapiro–Wilk test and the plots produced by the lm function were examined to ensure that datasets met the conditions required for the Pearson correlation test. Final plots were generated with ggscatter from the ggplot2 package (*Wickham, 2016*). Furthermore, phylogenetic independent contrasts were calculated using the function pic of the ape package (*Paradis & Schliep, 2019*) and similarly tested with Pearson correlation test, in order to assess whether correlations are maintained after correction for phylogeny. The correlation between genome size and each TE class was evaluated with Spearman's rank correlation test, and plots suggesting a linear relationship were further evaluated with a Pearson's test.

# RESULTS

## Mitochondrial genomes and phylogeny

Of the 32 mitogenomes for which assembly was performed, 29 of them could be assembled into a single circularized contig. Two species, *Umimayanthus chanpuru* and *Epizoanthus planus*, failed to generate a successful assembly. The processing of *Paleozoanthus reticulatus* resulted in a partial assembly of seven contigs, of which only four genes could be retrieved, *ATP6*, *ATP8* and *ND4L* on one contig (GenBank accession: OQ843460) and *COI* on another (OQ848443).

All other mitochondrial genomes were circularized and presented the complete gene set, displaying the same gene arrangement as described by *Chi & Johansen (2017)*; *COII*, *NAD4*, *NAD6*, *CYTB*, *COIII*, *COI* (with an intron), *NAD4L*, *ATP8*, *ATP6*, *NAD2* and *NAD5* including *NAD1* and *NAD3* gene copies in its intron. Mitochondrial genomes sizes ranged between 19,386 bp for *Epizoanthus rinbou* and 23,133 bp for *Umimayanthus parasiticus*. A

Peer J

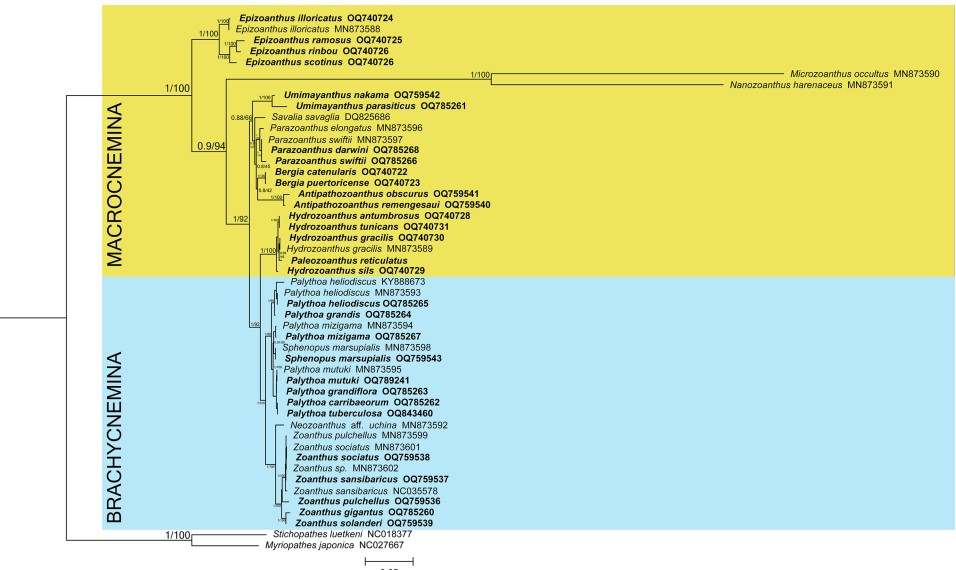

**Figure 1** **Bayesian inference phylogenetic tree of Zoantharia based on the concatenation of 13 mitochondrial protein-coding genes.** The phylogenetic trees computed with the Bayesian and the maximum-likelihood methods resulted in the same topologies, and hence node supports are displayed in posterior probabilities and bootstrap values. The two suborders in Zoantharia, Macrocnemina and Brachycnemina, are highlighted in yellow and light blue, respectively. Species names displayed in bold correspond to mitochondrial genome data added by the present research.

table summarizing the sizes of all complete mtgenomes is available in the supplementary material (Table S2).

The phylogenetic reconstructions performed with Bayesian inference and maximum-likelihood methods (Fig. 1) found the suborder Brachycnemina (highlighted in light blue) to be monophyletic with high support (Bayesian posterior probabilities = 1, maximum-likelihood bootstrap = 100%). Conversely, Macrocnemina (highlighted in yellow) was retrieved as paraphyletic, containing Brachycnemina as a sister clade of the macrocnemic genus *Hydrozoanthus*. Families Sphenopidae, including the genera *Palythoa* and *Sphenopus*, and Zoanthidae, comprising *Zoanthus* and *Neozoanthus*, were respectively found as monophyletic. The azooxanthellate, non-colonial species *Sphenopus marsupialis* was retrieved as a sister species to another azooxanthellate Sphenopidae, *Palythoa mizigama*. Similarly, *Hydrozoanthus* included a member of another genus, *Paleozoanthus reticulatus*, which was sister to *Hydrozoanthus gracilis*, with high support obtained only with the Bayesian inference (pp = 0.99; bootstrap = 66%).

## Genome sizes and repeated elements content

Genome sizes estimates were obtained for 18 species (Table S5). While estimates were obtained for *Epizoanthus planus* (38,964,917 bp) and *Paleozoanthus reticulatus* (28,412,256 bp), these were considered unreliable based on the spectrum generated by GenomeScope, which did not point to a clear *k*-mer peak. The additional 12 species for which genome sizes could not be estimated also showed profiles where GenomeScope "model did not

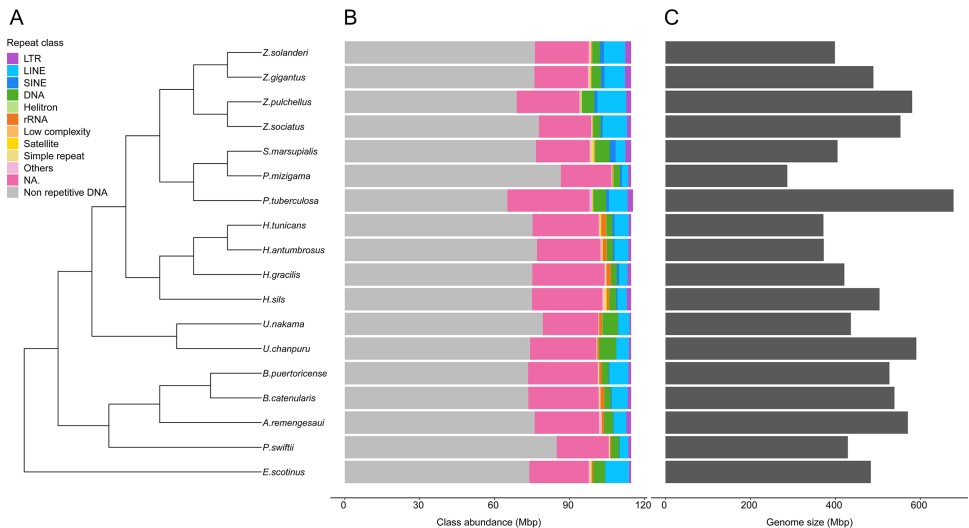

**Figure 2** **Phylogenetic relationships of 18 zoantharian species with their repeat class abundance and respective genome size.** (A) Cladogram of zoantharian phylogeny, (B) repeat class abundance, (C) genome size. "NA" refers to non-annotated repeated elements.

converge", pointing to the absence of a $k$-mer peak in the data. Genome size of zoantharians species ranged between 286 and 678 million base pairs (Mbp) (Fig. 2). The genera *Zoanthus*, *Umimayanthus* and *Hydrozoanthus* overlapped in range with genome sizes between 370 Mbp and 590 Mbp, and maximum genome size differences within genus of 160 Mbp. Genus *Palythoa*, however, comprised the maximum size differences at the scale of the order with a 2.4 fold variation and the maximum and minimum genome sizes, belonging respectively to *P. tuberculosa* and *P. mizigama* (Fig. 2C).

A range overlap in genome sizes between species in different genera was also apparent in the abundance of repeat reads, which accounted for 40 Mbp in several species (Figs. 2, S1). The read abundance for each repeated element class and species are reported in Table S6. Despite similar total repeat abundances, the proportions of repeat classes seemed to vary (Figs. 2, S1). Of all repeated elements, up to 30 Mbp (∼70% of repeated elements) could not be attributed to a known repeat class (Fig. 2). The abundance of unannotated repeats seemed to reach higher proportions in the comparatively smaller genomes of *P. mizigama*, *H. tunicans*, and *H. antumbrosus*. TEs were more abundant than other repeated elements. In particular, LINEs and DNA elements were consistently the most abundant classes among zoantharian species (Figs. 2, S1). LINEs elements were, in all species, especially represented by the LINE/L2 family and Penelope elements, which reached respectively up to 20,000 and 10,000 copies (Fig. 3, Table S7). LINE/RTE-BovB were particularly abundant in *Zoanthus* species, reaching about 15,000 copies in *Z. solanderi*, while being under 5,000 copies in other genera. Congeneric species of the genus *Bergia* appeared to have similar genome sizes of about 530 Mbp, and almost identical compositions of repeated elements. The same was true for *H. tunicans* and *H. antumbrosus,* which both had genome sizes of 370 Mbp. Conversely, species of *Umimayanthus* and *Zoanthus* showed a nearly identical
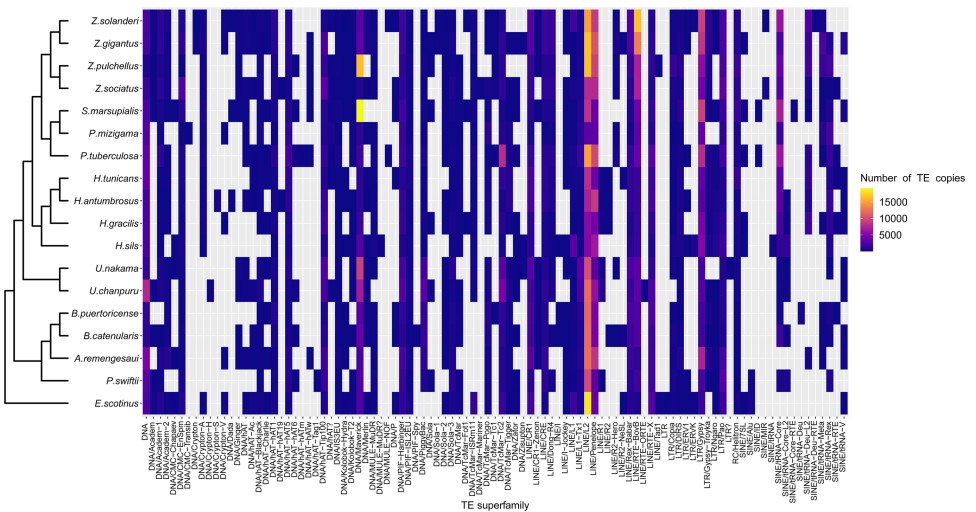

**Figure 3** **Heatmap representing transposable elements family abundance in 18 species of zoantharians.** TEs absent from a given species genomes are represented in cells with grey background.

composition of repeated elements despite having different genome sizes (Fig. 2). At a higher taxonomic level, there was no evident pattern of differences between species of the suborder Macrocnemina and Brachycnemina, except for the fact that macrocnemic zoantharians had a higher abundance of rRNA repeats. However, the clade including Brachycnemina and *Hydrozoanthus* appeared to have higher number of SINEs elements copies, while these were almost completely lacking from other macrocnemic zoantharians. *Sphenopus marsupialis* had a large amount of DNA/Maverick copies compared to other zoantharians (Fig. 3, Table S8).

Most of the transposable element landscapes showed a unimodal distribution with a spike of read abundance corresponding to a divergence of 0 to 2.5% from dnaPipeTE contig (Fig. 4). Abundance of TE reads increased gradually in *Zoanthus, Umimayanthus chanpuru* and *Palythoa tuberculosa*, while in other macrocnemic taxa, and in *S. marsupialis* and *P. mizigama,* most of the reads showed a peak at low divergences. DNA and LTR elements appeared to have a higher number of low divergence copies than LINEs in *S. marsupialis*. A few species displayed a bimodal distribution with increased number of LINEs at a high percentage of divergence. The second spike was stronger in *H. antumbrosus* which displayed an increased abundance of LINEs at a divergence of about 13%, while *H. tunicans,* its sister species according to the mitochondrial phylogeny (Fig. 1), did not show any other spike, and had relatively fewer LINEs at this degree of divergence. *Zoanthus solanderi* also displayed a small bump related to the activity of LINEs at ∼25% of divergence, and a similar bump was also present but much dampened in a close-related species, *Z. gigantus.* In most species, DNA elements were as abundant as LINEs at divergences higher than 2.5%. Conversely, in *Umimayanthus, Palythoa* and *Sphenopus* DNA elements appeared instead to be more important at divergences higher than 2.5%. At low divergences, LTRs appeared to have higher abundances, whereas SINEs disappeared, being at their peak abundance

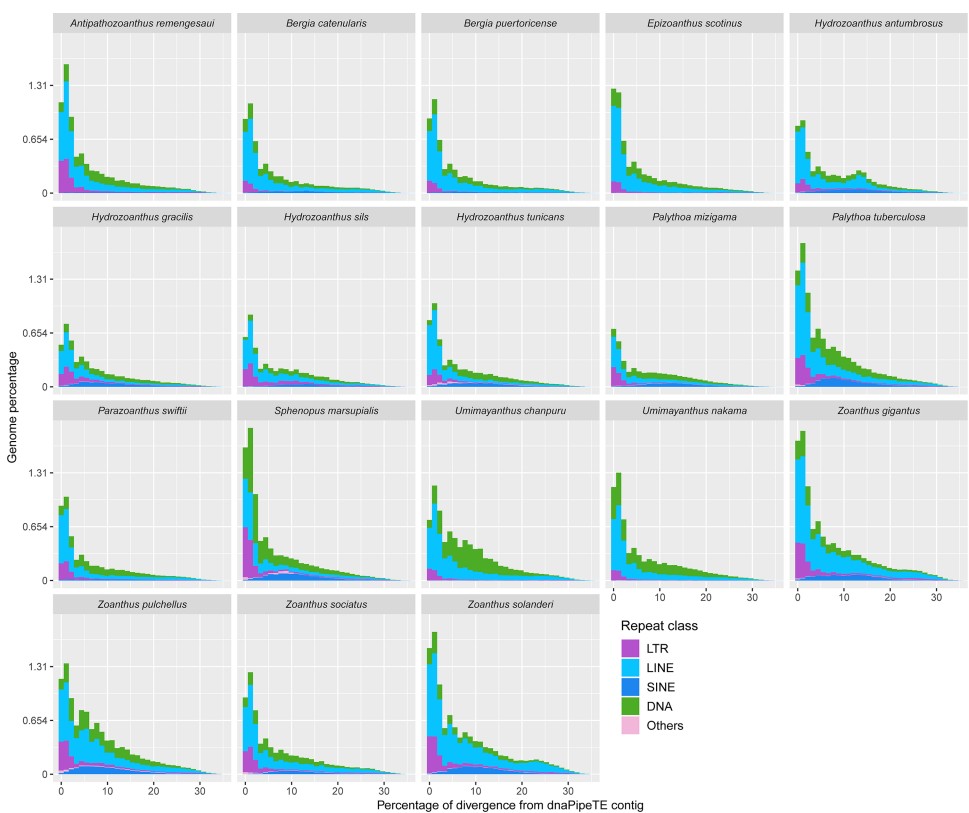

**Figure 4** **Transposable elements divergence landscapes for 18 species of zoantharians.**

(~0.12% of genome) at 10% divergence. Landscapes of the same 18 species including lower level of repeated DNA classifications are available in Fig. S2.

The repeated elements clustering in RepeatExplorer2 resulted in the analysis of 4,929,668 reads, of which ~60% were assigned to 340 superclusters, and 354 clusters. Total number of reads detected in each repeat class are summarized in Table S9. Many clusters were represented by all zoantharian species, in particular clusters displayed in Figs. 5 and S3 between cluster 349 and cluster 102, which were annotated as several different repeated element categories (45S, Maverick, LINEs and mobile elements). Other well-represented clusters among the zoantharian dataset were instead composed of unclassified elements, displayed between clusters 105 and 155 (Figs. 5, S3), which were found in increased abundance in *Zoanthus*. However, in general, clusters that were present among all zoantharian species did not seem to be found in high proportions with respects to genome size, as shown by their small repeat abundance on Fig. 5 (*e.g.*, clusters 13, 34, 343). Clusters retrieved in larger number were mostly species-specific or shared among closely related species of the same genus. In particular, several closely related species with almost identical genome sizes displayed very similar clusters in high abundance. This includes the two *Bergia* species with clusters 155 to 212, *Z. solanderi* and *Z. gigantus* (clusters 229 to 317), and the closely related *H. antumbrosus* and *H. tunicans*, with mostly satellites and LINEs (clusters 5 to 56). These groups of clusters corresponded essentially to satellite elements in

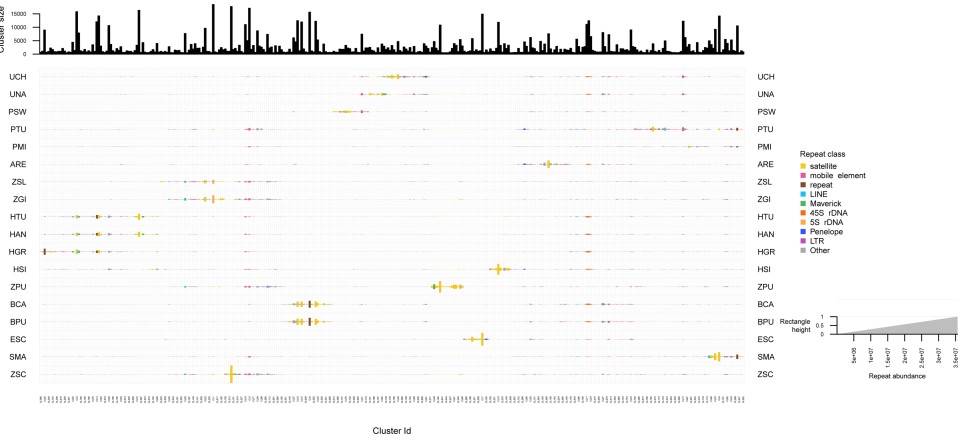

**Figure 5 Cluster sizes and annotations normalized by genome sizes among repeated elements of 18 zoantharian species.** Species names are shown as three letter codes. *U. chanpuru*: UCH; *U. nakama*: UNA; *P. swiftii*: PSW; *P. tuberculosa*: PTU; *P. mizigama*: PMI; *A. remengesaui*: ARE; *Z. solanderi*: ZSL; *Z. gigantus*: ZGI; *H. tunicans*: HTU; *H. antumbrosus*: HAN; *H. gracilis*: HGR; *H. sils*: I; *Z. pulchellus*: ZPU; *B. catenularis*: BCA; *B. puertoricense*: BPU; *E. scotinus*: ESC; *S. marsupialis*: SMA; *Z. sociatus*: ZSC.

the species pairs mentioned above. However, abundant clusters of LINEs were also shared among the two *H. antumbrosus* and *H. tunicans* (clusters 251 and 190) and among all *Zoanthus* species (cluster 29). 5S RNA was shared and particularly abundant in *Z. solanderi* and *Z. gigantus*. Conversely, several satellite clusters were found in high abundance in a single species only, mostly species displaying the highest genome size of their group (*H. sils*, *Z. pulchellus*, and *S. marsupialis*) (Figs. 2 and 5). *Z. pulchellus* and *Z. sociatus* had the highest genome sizes in *Zoanthus* (580 and 553 Mb respectively, Fig. 2) but had different clusters amplified; cluster 213 in *Z. sociatus* contained 35 million repeats while cluster 16 had 25 million repeats in *Z. pulchellus* (Fig. 5).

The screening of symbiont DNA in zoantharians datasets are shown in Fig. S4. Although overall, approximately 5% of reads mapped with symbiont genomes, for most species less than 1% of them mapped uniquely to a symbiont genome. In addition, the reads from several species associated with zooxanthellae did not map to Symbiodiniaceae genomes more than azooxanthellate zoantharians, suggesting low levels of contamination. The datasets where relatively high contamination levels were detected include *Zoanthus sansibaricus* (~9.43% of reads mapping singularly to *Cladocopium proliferum*), *Palythoa carribaeorum* (4.3%), *Palythoa grandis* (3.4%) and *Palythoa grandiflora* (4.3%), which were not included in main analyses.

## Correlation tests between genome size and repeated elements

Pearson's correlation test showed a correlation between the 18 genome sizes and the proportions of repeated elements, supported by an $R^2$ of 0.47 and a highly significant *p*-value of 0.0016. A weaker but statistically significant correlation was found between genome sizes and the percentage of transposable elements ($R^2 = 0.22$, $p = 0.05$), in which points appeared more dispersed (Figs. 6A, 6B). The tests of phylogenetic independent
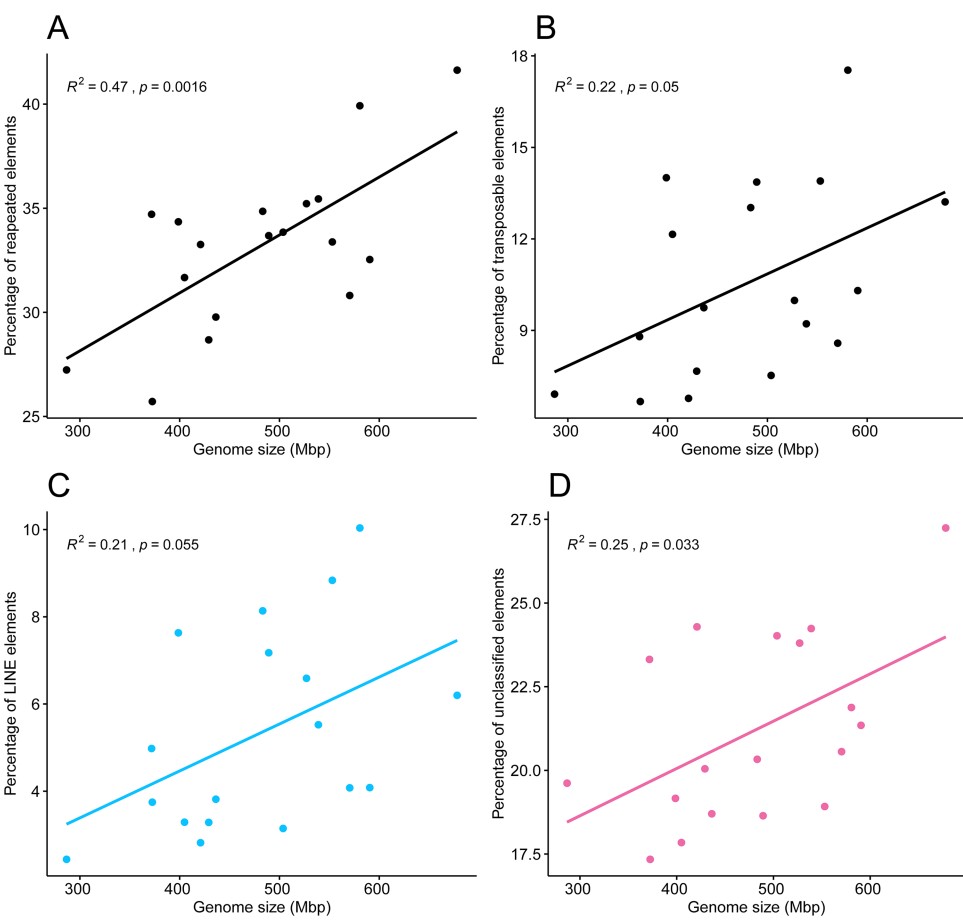

**Figure 6** **Pearson's correlation between genome size of 18 zoantharian species and their respective percentage among categories of repeated DNA.** Pearson's correlations between genome sizes and percentages of (A) total repeated elements, (B) transposable elements, (C) LINEs elements, and (D) unclassified repeats.

contrasts of genome sizes revealed significant relationships with the ones of same repeated element categories; repeated elements ($R^2 = 0.73$, $p = 1.3e-05$), transposable elements ($R^2 = 0.37$, $p = 0.0096$), LINEs ($R^2 = 0.28$, $p = 0.028$), and unclassified elements ($R^2 = 0.43$, $p = 0.0044$). All Pearson correlation tests were made under the assumption that residuals followed a normal distribution, which was confirmed by the Shapiro–Wilk test, with $p$-values > 0.05.

On the other hand, no significant correlation was noted by the Spearman correlation tests between genome sizes and each separate repeat class. Satellite elements, simple repeats, SINEs, rRNA, Low complexity elements, Helitrons, LTRs, and other repeats had no pattern of variation related to genome size.

## DISCUSSION

### Mitochondrial genomes and phylogeny of order Zoantharia

This study extended the datasets of zoantharian mitochondrial genomes compared to previous works *Poliseno et al. (2020)*, adding thirty additional mitochondrial genomes from twenty-two species and including four genera that had not previously been reported. Mitochondrial gene rearrangements have been reported in the close-related subclass Ceriantharia (tube anemones; *Stampar et al., 2019*) and in all other orders of Hexacorallia, including Actiniaria (sea anemones; *Johansen et al, 2021*), Corallimorpharia (corallimorpharians; *Lin et al., 2014*), and Scleractinia (stony corals; *Lin et al., 2014*), but none were observed here for Zoantharia. Similar to zoantharians, a lack of variation in gene orders in black corals (order Antipatharia) has also been noticed. However, sampling of 18 species of the group lead to the discovery of mitogenomic rearrangements, in the form of a loss of COI intron in two families (*Barrett et al., 2020*). The lack of evidence for gene rearrangements in zoantharians was also hypothesized to be due to the reduced sampling effort (*Poliseno et al., 2020*). Still, despite the increased taxon sampling of the present study, all mitochondrial genomes that could be completely assembled displayed the same gene order arrangement, which is identical to the one originally described by *Sinniger, Chevaldonné & Pawlowski (2007)* and *Chi & Johansen (2017)*. As of this study, Zoantharia remains the only hexacoral order without gene rearrangements in the mitochondrial genome. Although sequencing more species in the future may uncover different mitochondrial gene arrangements, the current situation suggests that biological factors may constrain the structure of mitochondrial genomes in zoantharians, as has been previously suggested for antipatharians (*Poliseno et al., 2020*). Our reconstructed mitogenomic phylogeny supports the position of suborder Brachycnemina as a clade within Macrocnemina coinciding with previous works (*Poliseno et al., 2020*). Therefore, Brachycnemina represents a paraphyletic group, with very high support both according to the Bayesian tree and the maximum-likelihood tree.

Even though the genome sequencing dataset of *Paleozoanthus reticulatus*, a specimen collected in 1982 (Table S1), showed signs of coverage issues with unreliable estimates of genome size, several mitochondrial genes could be retrieved from the sequencing data of this specimen. The specimen of *P. reticulatus* examined in this study is the only one reported since the species' original description in 1924 (*Kise et al., 2022*), and its phylogenetic position within the family Epizoanthidae has been unclear (*Kise et al., 2022*). Although *Paleozoanthus* is associated with the gastropod genus *Granulifusus,* similar to *Epizoanthus protoporos* (*Kise et al., 2022*), our molecular data suggest these species are not closely related. However, it has been previously suggested that this species might correspond to genus *Terrazoanthus,* in family Hydrozoanthidae, based on morphological features (*Low, Sinniger & Reimer, 2016*). Interestingly, the present phylogenetic reconstruction placed *Paleozoanthus reticulatus* within genus *Hydrozoanthus* (Fig. 1), which belongs to the same family as *Terrazoanthus* (*Kise, Maeda & Reimer, 2019*), Hydrozoanthidae. The phylogenetic placement of *Paleozoanthus reticulatus* within Hydrozoanthidae implies a previously undetected origin of symbioses with gastropods as members of this family are

generally associated with hydroids, octocorals, or bare substrate, while mollusc-associated zoantharians had only been confirmed until now from family Epizoanthidae (*Kise et al., 2022*; *Kise et al, 2023*). To clarify the phylogenetic position of *Paleozoanthus reticulatus,* including sequences of *Terrazoanthus* and other members of Hydrozoanthidae in future phylogenetic analyses is needed.

The present phylogeny also shows evidence of loss of symbiosis with Symbiodiniaceae within the family Sphenopidae, as the azooxanthellate species *Palythoa mizigama* and *Sphenopus marsupialis* were placed on internal branches within the primarily zooxanthellate genus *Palythoa*. This situation has been highlighted in previous phylogenies (*Dudoit et al., 2021*) and it has been suggested that the loss of photosymbiosis may even have occurred twice (*Irei, Sinniger & Reimer, 2015*). However, samples from another azooxanthellate species of this family, *Palythoa umbrosa*, are required to better clarify this point on the evolutionary history of photosymbiosis in Sphenopidae.

## Genome size of zoantharians and the role of the repeatome in their dynamics

This study presents the first genome size estimates for zoantharians. Many estimates of genome sizes across the order Zoantharia were within expected measures for most cnidarians, namely between 500 Mbp and 700 Mbp (*Adachi et al., 2017*). Among several genera of the order Zoantharia, genome sizes were found to overlap in their range (Fig. 2). For example, both genera *Zoanthus* and *Hydrozoanthus* included species with genome sizes of ∼350 Mbp and 500 Mbp. It is possible that this pattern reflects intraspecific variations; zoantharian species may have retained genome sizes constrained in a similar range yet exhibit fluctuations within this range. Large intraspecific variations have been documented in invertebrates, as in the extreme case of snapping shrimps, in which disparities up to 6 Gbp have been observed within one species (*Jeffery et al., 2016*). However, regarding cnidarians, the current knowledge points toward very narrow intervals; genome sizes are only known to vary up to 50 Mbp within jellyfish species *Sanderia malayensis* and *Rhopilema esculentum* (MD Santander, 2020, unpublished data) and less than 10 Mbp in anthozoans (*Adachi et al., 2017*). Alternatively, it seems more likely that different zoantharian groups have undergone complex evolutionary dynamic processes resulting in interspecific genome size disparities of similar amplitudes.

The present results suggest that repeated elements, and in particular transposable elements, are involved in genome size dynamics of zoantharians, explaining at least partly the variations observed. Indeed, observed genome sizes were correlated to the respective percentages of repeated and transposable elements (Figs. 6A and 6B). The paths to genome reduction or expansion are often the result of several processes, including transposable element activity or whole-genome duplication, which go in concert with changes in gene composition, genome structure and gene expression (*Martín-Durán et al., 2021*). Other lines of evidence are required to fully understand the processes surrounding genome size variations in zoantharians, in particular from species of *Palythoa* and *Zoanthus*, as these genera show signs of hybridization (*Reimer et al., 2007*; *Mizuyama, Masucci & Reimer, 2018*). However, the present results offer further insights into the contribution to genome
size of various repeated elements. Similar to what *Blommaert et al. (2019)* observed with rotifers, a diversity of repeated elements was found in the repeatome of zoantharians (Fig. 2). The annotation of repeated elements was challenging, as up to 70% of identified repeats could not be successfully annotated by dnaPipeTE (Fig. 2). Due to the difficulty of repeated element assembly and annotation, unclassified elements are expected. Although in some insect groups, unclassified elements only account for ∼10% of the total genome (*Goubert et al., 2015*; *Talla et al., 2017*), a study spanning several orders of Arthropoda showed a similar situation to our research, with more than 75% of repeats unclassified in some cases (*Petersen et al., 2019*). The number of unannotated repeats has also reached very high proportions in other cnidarians (*Xia et al., 2020*). Such results may reflect the scarce number of repeat references from cnidarians in databases, calling for more efforts in characterizing repeatomes of cnidarians. Additionally, the use of short-read sequencing may have contributed to the large amounts of unclassified repeats. However, annotation is likely the main explanation, as our assemblies' N50 and contig numbers (Table S10) were comparable to or better than those presented by the developers of dnaPipeTE (*Goubert et al., 2015*), who obtained significantly fewer unclassified elements.

Although we obtained large proportions of unclassified repeats in the dnaPipeTE analyses, the clustering and repeat annotation performed *via* RepeatExplorer2 suggested that they may be partly represented by satellite elements (Figs. 5, S3). Indeed, they accounted for ∼30% of the annotated elements in the comparative analysis (Table S9), yet they were almost absent from annotations *via* dnaPipeTE (Fig. 2). Conversely, numerous mobile elements could not be annotated from RepeatExplorer2. While this partly reflects the different sensitivities of the two pipelines and the databases that they use, the consistently high amounts of unclassified repeats in zoantharians highlight that much remains to be discovered with regards to their genomes. More efforts into assembling and characterizing their repeatomes will surely reveal interesting elements. Indeed, the percentages of unclassified repeat categories were found to be correlated to genome size (Fig. 6D), suggesting that elements with significance for genome size dynamics are contained among unclassified repeats.

Of the repeated elements that could be annotated, the most abundant classes were DNA transposons and LINEs. These results are in line with previous studies on the repeated DNA content of several cnidarians, where these two classes were also observed to be the most abundant (*Xia et al., 2020*).

The literature on the roles of repeated elements in genome sizes has largely focused on cases displaying extreme genome size variations. In these situations, dramatic changes of genome sizes in association with a single specific repeated class have been reported. Notably, the class of repeated elements involved varies between taxa; in larvaceans SINEs appear to drive genome size increases (*Naville et al., 2019*), while satellites and Helitrons were the main contributors in migratory locusts (*Shah, Hoffman & Schielzeth, 2020*). In *Hydra,* LINEs have had a major expansion event leading to dramatic genome size increase in the subgroup of brown *Hydra* (*Wong et al., 2019*). Although different repeated elements are clearly involved in genome size dynamics in different groups, the degree of variation between taxa is not well understood. The present dataset offers insights in this question

by adding to the knowledge of repeated elements in Cnidaria. Genome size variations in zoantharians do not appear to be as important as in *Hydra*, but still reach a maximum variation of 2.4 fold, between congeners *Palythoa tuberculosa* and *P. mizigama*. However, it is notable that LINEs–the class responsible for genome size expansion in *Hydra*– were consistently one of the most abundant in our dataset (Fig. 2B), and that a significant correlation between this class and genome size was detected (Fig. 6C). Furthermore, the repeat landscapes of most species showed a high number of LINEs with low divergence (Fig. 4). Such patterns have been interpreted as a sign of recent TE activity; TE copies in the genomes accumulate at a faster rate than mutations in their sequences (*Goubert et al., 2015*). Among them, two subfamilies appeared to be particularly abundant; namely LINE/L2 and Penelope elements (Figs. 4, S2). LINE/L2 were also one of the most abundant elements in the brown *Hydra* group (*Wong et al., 2019*) as well as *Aurelia* jellyfish (*Khalturin et al., 2019*). Therefore, this set of evidence suggests that the activity of various LINEs may have led to increased genome sizes in zoantharians, and potentially may have done so across Cnidaria. However, LINEs were found in varying, yet relatively small abundance overall and did not account for all the observed genome size variations on their own. Transposable elements are known to move other regions with them (*Langer et al., 2007*; *Qiu & Köhler, 2020*), and LINEs may have been transporting other regions of the genome with them, regions that could similarly account for the genome size discrepancies. In parallel with their effects on genome size, LINE/L2 may have impacted the evolution and functioning of zoantharians. Indeed, their role in the regulatory networks of housekeeping genes through the activity of LINE/L2-derived miRNAs have been demonstrated in humans (*Petri et al., 2019*).

Another seemingly important group of repeated elements in zoantharians are satellite elements, which represented the most numerous and largest clusters in the RepeatExplorer2 analysis (Figs. 5, S3). The comparative analysis performed *via* RepeatExplorer2 revealed instances of species-specific differentially expanded clusters. Closely related species (such as the pairs *H. antumbrosus* and *H. tunicans*, *B. catenularis* and *B. puertoricense*, *Z. solanderi* and *Z. gigantus*, Fig. 1) showed almost identical amplified clusters (Fig. 5, Fig. S3). On the contrary, species of those same genera but that branched earlier in the phylogenetic tree (Fig. 1) such as *H. sils*, or species that were simply more divergent, such as *Z. pulchellus* and *Z. sociatus*, showed unique cluster amplifications. These instances confirm that the species pairs mentioned above are very closely related, but also indicate that different satellite elements are amplified in the genomes of different species over the course of their evolution. Furthermore, this phylogenetic pattern is consistent with genome size dynamics. Indeed, several species that display large satellite elements clusters have larger genomes compared to other species of their group (*H. sils*, *Z. pulchellus*, and *S. marsupialis*, Fig. 2). In the migratory locust, expansion of satellite elements in the largest genomes were observed (*Shah, Hoffman & Schielzeth, 2020*). These authors suggested that rather of a causal relationship, the proliferation of satellites could be a consequence of genome expansion, as a mean to protect centromeric and telomeric chromosome regions after genome enlargement from transposable elements (*Shah, Hoffman & Schielzeth, 2020*). Considering their occurrence in species that have diverged for a long period of time,

this may also possibly be the case in zoantharians. However, the largest genome detected in this study, that of *P. tuberculosa,* did not display such large cluster amplifications of satellite elements. Considering our results on LINEs, unclassified and satellite elements, we conclude that the genome size patterns observed in zoantharians are likely the result of the activity of multiple groups of repeated elements.

## Traits potentially associated with genome size and repeated DNA

Two main evolutionary theories have been proposed to explain the puzzling variations observed in genome sizes; one that focuses on neutral processes and one on selective processes (*Blommaert, 2020*). In the first theory, the accumulation of DNA is considered a result of drift. The opposite theory suggests genome size is under the influence of selective forces and may impact organismal traits. In particular, genome size has been correlated to body size and egg size (*Naville et al., 2019*; *Stelzer et al., 2021b*), giving support to the nucleotypic hypothesis that proposes that genome size directly impacts phenotype by an effect on cell volume. However other traits have been suggested to potentially influence genome sizes, including geographical distribution (*Leinaas et al., 2016*), habitat (*Paule et al., 2021*) and effective population sizes (*Lefébure et al., 2017*). Although we did not formally analyze variations of genome sizes with phenotypic or biogeographic characteristics, a comparison with the phylogeny of zoantharians hints at features that may be affected. Symbiosis with Symbiodiniaceae dinoflagellates is one of the most studied facets of cnidarian biology because of its importance in sustaining the life of reef-building cnidarians and the subtropical to tropical ecosystem they support. This interaction is endosymbiotic and has large influence on host metabolism at the cellular level (*Davy, Allemand & Weis, 2012*), which in line with the nucleotypic hypothesis would have the potential to negatively impact genome size (*Adachi et al., 2017*). Because of this, a former investigation of genome sizes in Cnidaria attempted to find correlations between Symbiodiniaceae symbiosis and genome size (*Adachi et al., 2017*), but did not observe any significant relationship. However, in *Hydra,* genome size expansion has been associated with a switch away from symbiotic lifestyle (*Wong et al., 2019*). Indeed, the green hydra, with small genomes, maintains an obligate relationship with *Chlorella,* while symbiosis is not mandatory for strains of the brown *Hydra,* which have enlarged genomes (*Ishikawa et al., 2016*; *Wong et al., 2019*). In our study, on the other hand, a contrasting pattern was revealed between genome sizes and symbiosis in the group *Palythoa*. This genus comprised the largest genome size variation observed in all zoantharians—a 2.4 fold variation— between species with different symbiotic lifestyles. The maximum genome size was in zooxanthellate *P. tuberculosa*—678 Mbp, while the minimum size was in azooxanthellate *P. mizigama,* with 286 Mbp (Fig. 2). This makes *P. mizigama* within the range of the smallest cnidarian genomes recorded, that of *Sanderia malayensis* with a C-value of 0.26 pg, or about 250 Mbp (*Adachi et al., 2017*). Since macrocnemic zoantharians have similar ranges of genome sizes to brachycnemic zooxanthellate *Zoanthus* spp., it seems that the switch to a Symbiodiniaceae-associated lifestyle did not impact genome size. However, based on the *Palythoa* results, the loss of this relationship may be associated with smaller genome sizes. It can be hypothesized that the activity or loss of repeated DNA accompanying

genome size reduction of *P. mizigama* may have caused some genomic rearrangements impacting functions linked to symbiosis. As the loss of symbiosis may have occurred several times in *Palythoa* (*Irei, Sinniger & Reimer, 2015*), the rapid activity and movement of TE may be partly behind the apparent "switching on and off" of symbiosis in this group. Conversely, it is apparent that *P. tuberculosa* experienced genome enlargement. Following the reasoning of the nucleotypic hypothesis, genome sizes can be expected to be smaller in the case of a symbiotic organism, due to the symbiont effect on cell volume and metabolism (*Adachi et al., 2017*). However, symbiotic species may be subject to genome size increase through horizontal transfer and activity of transposable elements of their symbiotic counterpart. Although there is no documented evidence of TE transfer between Symbiodiniacae and hosts, transposable elements transcripts in *Symbiodinium* have been shown to be upregulated in situations of environmental stress (*Chen et al., 2018*). In addition, symbiotic *Symbiodinium* species have been found to have a larger content of transposable elements than free-living counterparts (*González-Pech et al., 2021*). Considering the abundance and activity of TEs in Symbiodiniacae, horizontal transfer may have contributed to the large genome size observed in the case of *P. tuberculosa.*

Alternatively, potential past events of hybridization may have contributed to the genome size variations observed in *Palythoa*. Hybridization is known to have occurred in zoantharians (*Reimer et al., 2007*) including genus *Palythoa* (*Mizuyama, Masucci & Reimer, 2018*). Hybridization is thought to potentially trigger the activation of TEs, leading to their accumulation in the hybrid genome (*Baack, Whitney & Rieseberg, 2005*; *Hénault et al., 2020*). This may have promoted species reproductive isolation as the increased transposition activity may have deleterious effects and cause sterility of the hybrids of two divergent populations (*Dion-Côté et al., 2014*; *Serrato-Capuchina & Matute, 2018*), and may have contributed to the evolution of *P. tuberculosa, P.* sp. yoron, *P. mutuki* and *P.* aff. *mutuki* (*Mizuyama, Masucci & Reimer, 2018*). Multiple aspects of zoantharian biology may be associated with genome size variations and transposable elements activity. To further understand the potential relationships between them, genome assemblies and estimates of genome sizes for other *Palythoa* species are necessary.

## CONCLUSIONS

In this article, we explored the relationships between phylogeny, genome size variations, and the repetitive elements composition of a scarcely studied group of cnidarians. Our results show that genome sizes observed in zoantharians are likely the product of complex historical dynamics of the repeatome. We found a high number of unknown repeats with potential implications in genome size. Recent expansion events of LINEs, DNA and satellite elements were identified in multiple species, raising questions on the role of these elements in genome evolution of cnidarians and the consequences of their activity. Until now no information was available for zoantharian genome sizes, and we here present such information for 18 specimens from five of the nine zoantharian families. This research demonstrates the power of next-generation sequencing projects aimed at understudied taxa, allowing a rapid increase in our basic understanding of such poorly studied groups. This

sequencing project also allowed us to clarify the phylogenetic position of *Paleozoanthus via* analyses of an old specimen; such work could very likely not have been performed utilizing traditional genetic methods. Finally, as there are notable questions related to the ecology, symbioses, development and evolution of zoantharians, the genome data and repeatome characteristics presented here will serve as important baseline data to investigate such questions in future genomic projects.

## ACKNOWLEDGEMENTS

We are grateful to NIG (National Institute of Genetics, Japan) for allowing us to use their supercomputer cluster for bioinformatic analyses, and in particular Tomohiro Hirai for their assistance in trouble shooting. We are also grateful to the ELIXIR-CZ project, which provided additional computational resources for the RepeatExplorer 2 analysis. We are thankful to Assoc. Prof. Takashi Nakamura, for his revision of a previous version of this manuscript and providing comments on the discussion.

### Funding

Funding was provided by Iridian Genomes, grant# IRGEN_RG_2021-1345 Genomic Studies of Eukaryotic Taxa. James Davis Reimer was supported by a JSPS Grant-in-Aid for Transformative Research Areas entitled "Environmental, ecological, and genetic observations of coral reef Symbiodiniaceae-host holobiont symbioses" (23H03821). Mylena Daiana Santander was supported by Coordenação de Aperfeiçoamento de Pessoal de Nível Superior 88882.377420/2019-01 and Maximiliano M Maronna by Fundação de Amparo à Pesquisa do Estado de São Paulo FAPESP 2016/04560-9 and FAPESP BEPE 2017/25907-0. Chloé Julie Loïs Fourreau received a scholarship from the MEXT (Ministry of Education, Culture, Sports, Science and Technology of Japan) and training workshop funding from a University of the Ryukyus ORCIDS grant. The funders had no role in study design, data collection and analysis, decision to publish, or preparation of the manuscript.

### Grant Disclosures

The following grant information was disclosed by the authors:
Iridian Genomes: IRGEN_RG_2021-1345.
JSPS Grant-in-Aid for Transformative Research Areas: 23H03821.
Coordenação de Aperfeiçoamento de Pessoal de Nível Superior: 88882.377420/2019-01.
Fundação de Amparo à Pesquisa do Estado de São Paulo: FAPESP 2016/04560-9, FAPESP BEPE 2017/25907-0.
MEXT (Ministry of Education, Culture, Sports, Science and Technology of Japan).
University of the Ryukyus ORCIDS.

### Competing Interests

James D. Reimer is an Academic Editor for PeerJ. Stacy Pirro is employed by Iridian Genomes.

## Author Contributions

- Chloé Julie Loïs Fourreau conceived and designed the experiments, performed the experiments, analyzed the data, prepared figures and/or tables, authored or reviewed drafts of the article, and approved the final draft.
- Hiroki Kise performed the experiments, analyzed the data, authored or reviewed drafts of the article, and approved the final draft.
- Mylena Daiana Santander conceived and designed the experiments, analyzed the data, authored or reviewed drafts of the article, and approved the final draft.
- Stacy Pirro performed the experiments, analyzed the data, authored or reviewed drafts of the article, and approved the final draft.
- Maximiliano M. Maronna conceived and designed the experiments, analyzed the data, authored or reviewed drafts of the article, and approved the final draft.
- Angelo Poliseno conceived and designed the experiments, performed the experiments, analyzed the data, authored or reviewed drafts of the article, and approved the final draft.
- Maria E.A. Santos analyzed the data, authored or reviewed drafts of the article, and approved the final draft.
- James Davis Reimer conceived and designed the experiments, analyzed the data, authored or reviewed drafts of the article, and approved the final draft.

## DNA Deposition

The following information was supplied regarding the deposition of DNA sequences:

The SRA and Genbank accession numbers for each presented sequencing data and mitochondrial genome are available in Genbank:

*Antipathozoanthus remengesaui*, PRJNA598175; *Antipathozoantus obscurus*, PRJNA598176; *Bergia catenularis*, PRJNA662740; *Bergia puertoricense*, PRJNA662980; *Epizoanthus illoricatus*, PRJNA598181; *Epizoanthus planus*, PRJNA676135; *Epizoanthus ramosus*, PRJNA598173; *Epizoanthus rinbou*, PRJNA662986; *Epizoanthus scotinus*, PRJNA662773; *Hydrozanthus tunicans*, PRJNA645597; *Hydrozoanthus antumbrosus*, PRJNA662983; *Hydrozoanthus gracilis*, PRJNA662988; *Hydrozoanthus sils*, PRJNA662735; *Paleozoanthus reticulatus*, PRJNA622546; *Palythoa carribaeorum*, PRJNA598184; *Palythoa grandiflora*, PRJNA598185; *Palythoa grandis*, PRJNA580275; *Palythoa heliodiscus*, PRJNA598194; *Palythoa mizigama*, PRJNA957836; *Palythoa mutuki*, PRJNA598187; *Palythoa tuberculosa*, PRJNA946699; *Parazoanthus swiftii*, PRJNA662982; *Parazoantus darwini*, PRJNA662981; *Sphenopus marsupialis*, PRJNA662993; *Umimayanthus chanpuru*, PRJNA662702; *Umimayanthus nakama*, PRJNA645598; *Umimayanthus parasiticus*, PRJNA662764; *Zoanthus gigantus*, PRJNA598193; *Zoanthus pulchellus*, PRJNA645596; *Zoanthus sociatus*, PRJNA598186; *Zoanthus solanderi*, PRJNA662769; *Zoanthus sansibaricus*, PRJNA598188.

## Data Availability

The raw data is available in the Supplementary Files.

## Supplemental Information

Supplemental information for this article can be found online at http://dx.doi.org/10.7717/peerj.16188#supplemental-information.

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
