# Peer review of "Genome sizes and repeatome evolution in zoantharians (Cnidaria: Hexacorallia: Zoantharia)"

_PeerJ, doi:10.7717/peerj.16188_

## Round 0.1 · original submission · Major Revisions

Thanks for submitting your manuscript, and congratulations on this important body of work from such a key group of organisms. As the reviewers pointed out, this is an excellent resource for cnidarian genomics, a fresh perspective beyond single organism genome analyses, and the importance of making this resource available to the scientific community. In general, the reviewers agreed on the quality of the data methodology presented and provided a constructive critique. I think the manuscript would be ready for publication after a reasonable round of edits. The reason I am sending this back for major revisions is to give you and the co-authors ample opportunity to address some of the concerns raised by Reviewer 1, with regards to the completeness of the data, the reason read analysis was favored over shallow-sequencing-draft assemblies, and potential contamination with symbiont DNA. Overall, the manuscript shows great promise and potential for advancing the field of cnidarian genomics. I look forward to seeing the revised version and working with you towards publication. Thank you again for your hard work and dedication to this important research. Please revise and resubmit the manuscript and address the comments raised during the review by all three reviewers accordingly.

Reviewer 1 ·

Basic reporting

See comments below

Experimental design

See comments below

Validity of the findings

See comments below

Additional comments

Summary of the paper and overall impression
In their manuscript “Genome sizes and repeatome evolution in zoantharians (Cnidaria: Hexacorallia: Zoantharia)”, the authors present whole genome sequencing data of various Zoantharia species and analyse these data to understand the evolution of repetitive elements in these organisms. These types of analyses can be challenging, but while the methods are sound and well described, I feel that more intensive analyses would better support the authors’ conclusions.

Main comments:
1. My main comment is on the choice of only using read-based analyses for genome size estimation and repeatome analysis. Both tools used rely on the assumptions that the sequencing libraries are complete, uncontaminated (see point 9) and unbiased, but there is little evidence presented in this manuscript to support either of these premises. Additionally, one sample was sequenced following PCR-enrichment of the sequencing library, which can cause an underrepresentation of GC-enriched reads in the sequencing output (also discussed in point 5). dnaPipeTE is great for comparing overall repeat content patterns across species, but is known to produce chimeric repeat sequences and, as the authors have highlighted, is susceptible to taxon-specific gaps in databases, meaning many repeats remain unannotated. In the discussion, the authors state “To further understand the potential relationships between them, genome assemblies and estimates of genome sizes for other Palythoa species are necessary”, but this left me wondering why no genome assemblies were reported in this work. Unless I understood wrong, many of the sequenced specimens had over 15x coverage, which is sufficient to produce a draft assembly, which could then be subjected to completeness checks (such as BUSCO), other genome size estimate approaches, and more in depth repeat element annotation and curation. Pipelines like panEDTA, for example, are great for annotating and curating pan-species transposon libraries from genome assemblies. Of course an approach like this will still miss some repeats, but multiple analyses (assembly based, coverage analyses, read based, etc) can be combined to make a much stronger conclusion.
2. I found the introduction a bit lacking in detail in parts, and the three topics (repeat elements, genome size, and zoantharian evolution) to be a little bit disjointed. These could be better tied together throughout. The first sentence (“In biology, the ideas that some organisms are more complex than others and that evolution is directed towards progress and increasing complexity have oriented many research topics.”) seems to suggest that evolutionary biologists support the idea that evolution has some kind of direction, when it is simply the process by which natural selection and genetic drift influence the inheritance of genes. The authors also suggest that this thinking forms the basis of the C-value paradox, while the C-value paradox does not state anything about the direction of evolutionary change, simply that there is a lack of correlation between genome size and organismal complexity.
3. L121-123: “This is also true for the study of repeatome and genome sizes, as many groups still lack basic genome size information.” This statement would benefit from some details (e.g. numbers of species in databases, known genome size ranges, etc) and any relevant references.
4. For the mitochondrial genome assemblies, I would be curious if other programmes (e.g. getOrganelle) would be successful in assembling the samples that failed. I would also be curious as to what kinds of sequences are responsible for the differences in mitochondrial genome size even where the gene arrangement is conserved.
5. This point is already alluded to in point 1- The DNA extraction and sequencing methods were different for P. mizigama as compared to every other sample sequenced in this study, namely this DNA went through a round of PCR enrichment. PCR can introduce a GC-bias in sequencing libraries, and this and lead to exclusion of regions of the genome that are particularly GC-enriched. Considering the smaller genome size estimate of this sample, this should be considered, and GC profiles of all sequencing libraries compared and discussed. This does not necessarily invalidate this result, but is worth extra scrutiny and discussion.
6. The description of the phylogeny (Lines 351-355) is not overly clear to me and could use extensive clarification. Additionally, the figure legend mentions node supports, but there are no node support values in the figure. The figure legend also does not explain why some names are in bold and others are not.
7. It is not clear to me if correlations between genome size and content were phylogenetically corrected. If not, this should be included. If this was done, it needs to be clarified in the methods.
8. On lines 580-581 the authors suggest “that elements with significance for genome size dynamics are contained among unclassified repeats”, it would be interesting to identify the most important unclassified repeats and attempt to further classify or identify them through more in depth analyses such as trying to identify functional domains in these elements and classifying them according to Wicker, 2017.
9. The authors extensively discuss the potential impacts of symbiosis on genome size, but there appear to have been no steps taken to consider the impact of symbiosis on genome sequencing results- how do we know the symbiont DNA has not been included?
10. Table 1 contains a lot of information I would consider supplementary as well as not clearly providing an overview of all samples in this study. I would suggest producing a map indicating the collection location of each sample (including those already reported in other studies) and additionally producing a table for the manuscript which includes the sample name, location, and number of Gbp of sequencing (not number of reads) produced per sample and attach the current version of Table 1 as a supplement. Please also check the number of samples in the table/figure in comparison to those reported in the text, these numbers are currently different.


Minor comments:
1. L196-197 “Before the mitochondrial assembly, paired-end reads adapter sequences were removed in Trimmomatic v. 0.39 with default parameters (Bolger et al., 2014).” This seems like it belongs in the “Sampling and Sequencing” section of the methods
2. Lines 371 and 372: In my opinion, the phrasing including “maximum disparities” is not a very clear way of explaining these results.
3. Lines 433-435: “However, in general, clusters that were present among all zoantharian species did not seem to be found in high proportions with respects to genome size (Fig. 5).” The statement itself is fine, but I do not feel that Figure 5 supports this so I am unsure why it is referenced here.
4. In the correlation between genome size and transposable element content (reported on Lines 455-457 and Figure 6), did this include ALL transposable elements, or only those able to be classified?
5. Line 524: “This study presents the first genome size measurements for zoantharians”, these were not genome size measurements, but estimates.
6. Line 651-654: “However other traits have been suggested to potentially be impacted by genome sizes, including geographical distribution (Leinaas et al., 2016), habitat (Paule et al., 2021) and effective population sizes (Lefébure et al., 2017).” These are traits that are typically thought to influence genome size, not to be impacted by genome size.
7. I would also consider Table 2 to be supplementary information
8. Figure 2: Part B of this figure and Supplementary Figure 1 appear identical. This redundancy is not required. If they are different, please clarify. Additionally, the differences in repeat elements in each genome would be more clear and intuitive if the section “non repetitive DNA” was removed from the stacked bars. Please also clarify the meaning of the class “NA”. The authors may also consider highlighting interesting differences in genome size and content in this figure as they do in the text.
9. I would find Figure 3 more informative if genome size estimates were included along with the species labels, or if species were arranged by genome size.
10. General comment on all figures: please check resolution of figures, text size (especially Figure 5), consistent themes (background colour, etc), and that colour palettes are colour blind friendly.
11. Please check that all references are present and in the correct order in the reference list

·

Basic reporting

Fourreau and colleagues generated genome data for 32 zoantharians, a group of understudied cniadiarians, to study their mitochondrial genomes and the contributions of repetitive elements to genome evolution. The study presents new insights for this group of cnidarians and provide valuable data for future research. The article is mostly well written, enough background is provided, methods and results are clear and the visuals (figures and tables) are well suited to support their findings. All generated data has been made available to the public. The following are minor recommendations to the authors:

ABSTRACT
- L69-61: Consider replacing "attributed to" with "annotated" or something of the sort.
- L151: "ezRAD"

MATERIAL AND METHODS
*Several parts of these section provide too much detail. In my opinion, it could be written in a more succinct manner. Some of the bullet points below show examples of this, but the author might want to pay a more thorough revision to this section.
- L240-4: Instead of mentioning all the commands used to generate k-mers and the corresponding frequency histograms, the authors could simply mention that they used Jellyfish for these purposes.
- L281-5: This paragraph could be removed.
- L324-6: This sentence could be removed. Its content could be incorporated within the next sentence with a few extra words.

RESULTS
- L347-51: This paragraph reads like methods.
- L375: Consider referencing Fig. 2 in a sentence above before introducing directly Fig. 2C.
- L377: Did the authors mean 400 Mbp instead of 40?

DISCUSSION
- L505: Hydrozoanthidae should not be italicized.
- L617-9: This sentences does not relate to Fig. 6, consider removing this reference.
- L687: Space missing in P. tuberculosa.
- L697: This sentence reads incomplete; contributed to what?

REFERENCES:
- L761: Aranda instead of Arand.
- L915-36: The references starting with 'S' are not fully alphabetically sorted and the "Shat et al. 2020" reference is missing in the list.

TABLES:
- Table 2 could be moved to the supplementaries.

FIGURES
- Fig. 1: No node supports are visible in the figure.
- Fig. 2: Clarify that "NA" stands for "not annotated".
- Fig. 5: Cluster labels are unreadable. Consider moving this figure ("the full picture") to supplementaries and showing only panels of interesting clusters (e.g. most represented amongst all species).

Experimental design

The subject of the study falls well within the scope of PeerJ. The research questions and objectives of the investigation, as well as the knowledge gaps to address, are clearly presented. The methods used are sound and presented with extensive detail. Minor methodological details to clarify:

- L176: Please, provide full name of sequencing platform. Is it Illumina HiSeq 2500?
- L187: Same as above.
- L257-60: The parameters specified for Trimmomatic do not match what is written in the text.

Validity of the findings

Results and interpretations are well supported by the data and methods employed. Statements in discussion and conclusion are well sustained and of relevance to the study. Minor points to consider for the discussion:

- L453-68: To me, R-squared gives a better indication of correlation strength than R because it shows the variation in the dependent variable that can be related to the independent variable. Therefore, R = to 0.73 (or R-squared = 0.53) is not a very strong correlation.
- L587-616: Although the author discuss how TEs have impacted genome evolution in other systems, that could be of relevance to the surveyed zoantharians, they fail to mention other potential mechanisms. Their findings suggest a correlation between genome size an certain mobile elements, yet the fractions that these elements occupy in their genomes is small. It is possible, that mobile elements moving around additional regions of the genome when jumping around. Of course, (nuclear) whole-genome assemblies are required to asses what has been moved around through mobile elements, which emphasizes the need for additional high-quality genome data for this group.

Additional comments

The following points would make the study even more interesting but are not required:

- The fraction of reads that went into the mitochondrial assemblies out of the total number of reads generated per sample.
- Since the authors framed the issues regarding zoantharians phylogenetics and systematics, it would have been exciting to see further phylogenomic analyses feasible with their data other than the traditional multi-gene phylogeny. For instance, they could have done k-mer based phylogenomics with the mitochondrial assemblies (Bernard et al. 2021, Bacterial Pangenomics: Methods and Protocols), that can also been implemented with repetitive elements (Lo et al. 2022, Frontiers in Plant Science).

Reviewer 3 ·

Basic reporting

The authors fill an important gap in genome size and repeat analyses by sequencing and analyzing the genomes of a diverse set of species within the Zoantharia order. They successfully demonstrate the complex relationship between genome size and TE content, as well as some consistent patterns within. In addition, this work also highlights the need for TE curation efforts to include a wider range of species.

I am very impressed by the level of detail shared in the methods as well as the data generated and stored. These authors should be used as an example of how data should be presented for reproducibility purposes. This is made all the more stunnnig due to the sheer amount of information generated. I cannot adequately emphasize how important this is from the perspective of a reviewer and researcher.

I approve this manuscript with minor revisions (see below).

Introduction:
While the introduction is well-written and a nice balance of detailed and concise, given the amount of information provided within, it appears under-cited, particularly in the background of Tes.

Line 82: given the rapid rate at which species are sequenced and analyzed, the addition of a more recent citation compared to 1950 is due.

Line 134: Is “Rafinesque, 1815” supposed to be a citation?

Methods:
The methods are quite thorough and show a great deal of forethought and planning.

Line 217: How were the mitochondrial genes retrieved? Is this solely based on the mitochondrial genome assembly and annotation described in the manuscript, or were other databases also utilized? Was a BLAST search performed? Please add enough detail that others can find the genes you describe.

Lines 303-304: Are there any details you can add to provide the audience some clarity as to how you performed your manual curation, or briefly describe some of the methods in the cited paper?

Results:
Line 340: Is this a new paragraph, or should it be joined with the previous?

Line 351: Does this AIC model fit with the data based on your understanding of the genes? Please provide rationale for the audience for why this is a good fit.

Lines 352-363: Based on the descriptive text here for Figure 1, I would highly recommend additional clear taxonomic labeling to the aforementioned figure to enhance audience understanding.

Line 369: Why couldn’t the genomic size data be obtained for these additional 12? Is it the same rationale for the two mentioned species? Please provide the rationale.

Figures:
Figure 1: I do not see the node support on this figure. In addition, it is not clear based on the labeling which species belong to Brachycnemina or Macrocnemina. Please adjust the labeling for reader clarity.

Figures 3: Could you please provide the same phylogenetic tree on the bottom of the figure near the species names as you did in Figure 2? It might provide some clarity for the audience.

Experimental design

No comment

Validity of the findings

The methods are quite thorough and show a great deal of forethought and planning. I have the following comments to enhance the manuscript:

Line 217: How were the mitochondrial genes retrieved? Is this solely based on the mitochondrial genome assembly and annotation described in the manuscript, or were other databases also utilized? Was a BLAST search performed? Please add enough detail that others can find the genes you describe.

Lines 303-304: Are there any details you can add to provide the audience some clarity as to how you performed your manual curation, or briefly describe some of the methods in the cited paper?

The conclusions and discussion are well-written and an in-depth analysis of the data, considering all possible angles.

---

## Round 0.2 · Minor Revisions

Thanks for revising the manuscript promptly and efficiently. The reviewers agree that most of the major issues have been addressed adequately and that the manuscript is mostly ready. I am sending the manuscript back for minor revisions to allow you to address the minor issues raised in this round of review, so the final version is ready for publication.

Reviewer 1 ·

Basic reporting

Overall, I find that the authors have appropriately responded to all reviewer comments. However, in my original review report, I pointed out that not all references were in the reference list at the end and in the correct order, and there is at least one reference still missing (Blommaert, et al, 2019), so I highly encourage the authors to once again carefully check the referencing of this manuscript. Additionally, I could not find the bioproject of P. tuberculosa (PRJNA946699) on NCBI, and the SRA entry (SRR23916682) was not public yet. The authors should make sure this will be in order by the time the manuscript is published.

Experimental design

The authors have done well to address the comments in this section. However, I have some feedback for future work regarding contaminants. The level of contamination in the analysed samples here was not high, but in future it would be best practise to remove contaminant reads prior to anaylses.

Validity of the findings

No comment

Additional comments

Well done on these analyses and responding to all comments thoroughly and professionally!

·

Basic reporting

I endorse publication of this article almost as it is. I only have minor suggested changes that I 11 have overlooked in my previous review.

- IMO, an R2 = 0.4 correlation is not a strong correlation. I would encourage the authors to modify the text accordingly throughout. This is my major concern.
- L49: The "x" for coverage is translated as times. Therefore, having "x" and "fold" is redundant. Please, choose one and keep it consistent throughout. In fact, instead of an "x" the correct symbol is the multiplication sign.
- L102: "LINE elements" is redundant. I feel like in general the word "elements" is somewhat abused in the text, e.g., both LINE and satellite "elements" could be mentioned just as LINEs and satellites. Please, consider and revise throughout.
- L119-21: Sentence oddly phrase; there might be an "and" missing?
- L172: Space missing.
- L185: To me, "on an Illumina..." sounds more appropriate.
- L186: Remove space at the beginning of the line. Also, add the SRA acronym after spelling it out.
- L190: Add FastQC version.
- L196: The k in k-mer should be in italics; please, correct throughout. Make sure gene symbols are in italics as well.
- L196-200: Please, specify which sequences (including GenBank accessions) were used as seed/reference for each of the genomes, maybe in a supplementary table.
- L224: You could introduce the ML acronym here and use it throughout the text.
- L237-9: Fix indent.
- L260-4: The Shoguchi et al. (2018) study only generated a Cladocopium genome, the other one is probably the one from Liu et al. (2018), originally published as a C. goreaui genome, but this species has been named C. proliferum (Butler et al. 2023, Journal of Phycology). The Durusdinium genome corresponds to Durusdinium trenchii.
- L355: "Table S3".
- L368: "overlap".
- L373-5: This is not what can be observed from Fig. 2.
- L413-432: It is not possible to read the cluster names in Fig. 5. Likewise, it is hard to tell the type of repeat element.
- L442: Replace C. goreaui with C. proliferum.
- L622-4: Sentence reads odd and could be tightened up.
- L669-73: The authors might consider including in this part of their discussion that a higher abundance of TEs are expected in symbiotic Symbiodiniaceae (González-Pech et al. 2019, Trends in Ecology & Evolution) and that preliminary evidence of this has been found in Symbiodinium genomes (González-Pech et al. 2012, BMC Biology).

Experimental design

No comments.

Validity of the findings

No comments.

Additional comments

No comments.

Reviewer 3 ·

Basic reporting

No comment.

Experimental design

No comment.

Validity of the findings

No comment.

Additional comments

The authors have adequately responded to all of my concerns. I fully support the publication of this manuscript.

---

## Round 0.3 · accepted · Accept

Thanks for making the minor corrections from the last round of review. The corrections are sufficient and have addressed all the minor requests by the reviewers. The manuscript is now ready for publication. Congratulations!